

# Microstructure-based simulations of the viscous densification of snow and firn

Kévin Fourteau[1,a], Johannes Freitag[2], Mika Malinen[3], and Henning Löwe[1]

[1]WSL Institute for Snow and Avalanche Research SLF, Flüelastr. 11, 7260 Davos, Switzerland
[2]Alfred-Wegener-Institut, Helmholtz-Zentrum für Polar- und Meeresforschung, Bremerhaven, Germany
[3]CSC – IT Center for Science Ltd., Espoo, Finland
[a]Present address: Univ. Grenoble Alpes, Université de Toulouse, Météo-France, CNRS, CNRM, Centre d'Études de la Neige, Grenoble, France

**Correspondence:** Henning Löwe (loewe@slf.ch)

**Abstract.** Accurate models for the viscous densification of snow and firn under mechanical stress are of primary importance for various applications, including avalanche prediction and the interpretation of ice cores. Formulations of snow and firn compaction in models are still largely empirical, instead of using microstructures from micro-computed tomography to numerically compute the mechanical behavior directly from the physics at the micro-scale. The main difficulty of the latter approach is the
choice of the correct rheology/constitutive law governing the deformation of the ice matrix, which is still controversially discussed. Being aware of these uncertainties, we conducted a first systematic attempt of microstructure-based modeling of snow and firn compaction. We employed the Finite Element suite ElmerFEM using snow and firn microstructures from different sites in the Alps and Antarctica to explore which ice rheologies are able to reproduce observations. We thereby extended the ParStokes solver in ElmerFEM to facilitate parallel computing of transverse isotropic material laws for monocrystalline ice.
We found that firn (density above $550\,\mathrm{kg\,m^{-3}}$) can be reasonably well simulated across different sites assuming a polycrystalline rheology (Glen's law) that is traditionally used in glacier or ice sheet modeling. In contrast, for snow (density below $550\,\mathrm{kg\,m^{-3}}$) the observations are in contradition with this rheology. To further comprehend this finding, we conducted a sensitivity study on different ice rheologies. None of the material models is able to explain the observed high compactive viscosity of depth hoar compared to rounded grains having the same density. While on one hand our results re-emphasize the limitations
of our current mechanical understanding of the ice in snow, they constitute on the other hand a confirmation of the common picture of firn densification as a foam of polycrystalline ice through microstructure-based simulations.

## 1 Introduction

Once deposited on the ground, snow and firn are subject to the overburden stress imposed by the weight of the overlying snow or firn column. This causes the slow and viscous compaction of the snow and firn layers and their densification over time
(e.g. Kojima, 1975; Herron and Langway, 1980). Accurate prediction of the rate of the compaction is of primary importance for various applications of the cryospheric sciences. For instance, the densification of firn determines the time scale at which atmospheric gases are enclosed in polar ice (Schwander et al., 1993). Faithful modeling of firn densification is thus a critical



component for the interpretation of ice cores (Goujon et al., 2003; Witrant et al., 2012; Buizert, 2021). However, observed variations in the densification rate of different layers still lack a conclusive explanation in view of either microstructural or

compositional origins (Hörhold et al., 2012; Fujita et al., 2016). This situation is in remarkable similarity to snow. The slow densification of seasonal snow determines the time scale for the existence of persistent weak layers as a major source of avalanche danger (Schweizer and Lütschg, 2001). Faithful modeling of snow densification would greatly improve the capacity of a detailed snowpack model to predict this time scale and the presence of persistent weak layers over the winter season. Still, for snow there are well known differences in densification rates between different layers (rounded grains and depth hoar)

(Kojima, 1975; Calonne et al., 2020) that still lack a conclusive explanation. In view of these similarities, holistic approaches to snow and firn densification are desired to identify common microstructural controls.

Due to the progress in micro-computed tomography ($\mu$CT) in cryospheric sciences within the last decade, 3D microstructures of snow and firn samples are now widely used (e.g. Löwe et al., 2013; Calonne et al., 2019; Letcher et al., 2022). Based on $\mu$CT, the effective material properties of snow or firn can now be derived in a physically consistent manner, based on the

underlying microstructure and the physics at the pore scale. This leads to an effective (upscaled) material characterization using homogenization methods and numerical simulations (Torquato, 2002; Auriault et al., 2009). For example, for the thermal conductivity of snow and firn, upscaling methods have been widely used, yielding a fairly consistent picture of how the microstructure determines the effective thermal properties (Calonne et al., 2011; Löwe et al., 2013; Riche and Schneebeli, 2013; Calonne et al., 2019; Fourteau et al., 2021). For snow and firn compaction, such a consistent picture is still lacking.

Despite the aforementioned requirement on accuracy, the formulation of slow compaction of snow and firn in models remains largely empirical (e.g. Vionnet et al., 2012; Lundin et al., 2017) and lacks a detailed microstructure-based justification. To the best of our knowledge, only Theile et al. (2011), Chandel et al. (2014), and Wautier et al. (2017) have made the attempt so far to estimate the effective viscous response of snow using $\mu$CT images and microstructure-based non-elastic (plastic or viscoplastic) simulations. While all of them led to an apparent agreement with measurements, they were based on very different

assumptions about the rheology of the ice material accommodating the deformation at the micro-scale. The main difficulty for developing a microstructure-based formulation of compaction and its integration in snow and firn models, is the present disagreement concerning the dominating mechanism(s) driving the mechanical deformation of ice at the micro-scale. While it is generally accepted that firn densification occurs through creep in the ice matrix, several types of creep have been proposed in the literature. For instance, Arthern et al. (2010) assumes the ice to deform according to a Nabarro-Herring creep, where

deformation occurs through vacancy diffusion in the crystals, while Barnola et al. (1991) or Salamatin et al. (2009) assume a power-law creep law that originates from dislocation creep. For snow, the mechanism of deformation remains even more elusive. Some studies support the idea that the ice in snow deforms through a dislocation-creep mechanism (Kirchner et al., 2001; Scapozza and Bartelt, 2003; Wautier et al., 2017). This led to the appealing concept of snow as foam of (polycrystalline) ice (Kirchner et al., 2001). Other studies rather support the idea that the compaction of snow involves grain boundary sliding

(Alley, 1987; Salamatin et al., 2009; Schultz et al., 2022). This uncertainty is for instance reflected by the choice of Wautier et al. (2017) considering three different ice power-law rheologies with an ice fluidity about 1000 times higher than the usually





reported value (e.g. in Chap. 3 of Cuffey and Paterson, 2010). Therefore, the most promising approach is to allow for flexibility in the material model when comparing simulated densification rates with observations using diverse data sets.

The aim of this article is to study if the viscous compaction of snow and firn could be consistently simulated directly

from the microstructure using a similar numerical approach. We use different microstructures (rounded snow, faceted snow, and dense firn) to assess the validity of this modeling approach over a broad range of possible microstructures. Snow and firn microstructures were obtained by $\mu$CT imaging of snow and firn cores from different sites in the Alps and Antarctica, where independent estimates of observed densification rates are available through complementary measurements. We employ and extend ElmerFEM as computational platform to make contact to established procedures in the ice flow community. In

this setup, we investigate which viscous ice matrix rheology (variants of Glen's law) is able, or is not able to appropriately reproduce the observations of snow and firn compaction.

The manuscript is organized as follows. Section 2 presents the minimal theoretical background used throughout this work. Section 3 details the data and numerical simulations set-up that were used for the study, and Section 4 presents and discusses our results.

## 70 2 Theoretical background

For snowpack and firn models, it is still impossible to represent the 3D microstructure of a whole snowpack or firn column explicitly. Instead, these models rely on an upscaled (or volume averged) description where the snow/firn can be described as a homogeneous medium characterized by effective material properties (Torquato, 2002; Auriault et al., 2009) that are often referred to as *macroscopic* to emphasize the separation of scales from the characteristic (microscopic) length scales where the

theoretical description is formulated. Accordingly, the compaction of snow and firn is described in terms of a macroscopic strain rate $\dot{E}$ resulting from the macroscopic overburden stress $\Sigma$. For snow or firn modelling, purposes it is thus required to derive macroscopic constitutive laws $\dot{E}$ as $\dot{E} = f(\Sigma)$, where $f$ is a function that characterizes the mechanical response of snow/firn that depends on the snow/firn microstructure and other relevant variables, such as the temperature (Arnaud et al., 2000; Bartelt and Lehning, 2002; Vionnet et al., 2012). The aim of this section is to provide some general considerations on

the functional form of $f$.

### 2.1 Conservation laws

We start from the equations governing the mechanical response at the micro-scale. As we are interested in the slow compaction of snow and firn over timescales of days to years, we resort to the common assumption that elastic stresses have been relaxed and that the ice in the microstructure is responding in a purely viscous fashion. This assumption is supported by the relaxation

time scale of elastic stresses in snow, that is of the order of hours (Theile et al., 2011). Technically, this assumption allows to describe the ice as an incompressible, (non-linear) viscous fluid and the deformation of the viscous ice matrix is then governed



by the Stokes equations

$$\nabla \cdot \boldsymbol{\sigma}(\boldsymbol{x}) \;=\; 0 \quad \boldsymbol{x} \in \Omega_{\mathrm{i}} \tag{1}$$

$$\nabla \cdot \boldsymbol{u}(\boldsymbol{x}) \;=\; 0 \quad \boldsymbol{x} \in \Omega_{\mathrm{i}} \tag{2}$$

where $\boldsymbol{x}$ is the three dimensional position vector, $\boldsymbol{\sigma}(\boldsymbol{x})$ is the microscopic stress tensor field, $\boldsymbol{u}(\boldsymbol{x})$ is the microscopic deformation velocity field, $\Omega_{\mathrm{i}}$ is the three-dimensional domain that is occupied by the ice matrix, and $\nabla\cdot$ denotes the divergence operator. The Stokes equations need to be complemented by a constitutive law that characterizes the rheology of the ice.

## 2.2 Constitutive law

The constitutive laws considered in this work are variants of Glen's law which involve a power-law non-linearity of the form

$$\dot{\boldsymbol{\epsilon}}(\boldsymbol{x}) = s_{\mathrm{eq}}^{n-1}\boldsymbol{a}(\boldsymbol{x},T) : \boldsymbol{s} \; . \tag{3}$$

Here $\dot{\boldsymbol{\epsilon}}(\boldsymbol{x})$ is the microscopic strain rate tensor, $\boldsymbol{s}$ is the deviatoric part of the microscopic stress tensor $\boldsymbol{\sigma}$, $s_{\mathrm{eq}} = \sqrt{s_{ij}s_{ij}}$ is the equivalent deviatoric stress, $n$ is a non-linearity exponent, and $\boldsymbol{a}$ is referred to as the fluidity. In the general case, the fluidity $\boldsymbol{a}$ is a fourth-order tensor that depends on temperature $T$ and potentially on the position $\boldsymbol{x}$ in the microstructure, if e.g. the c-axis orientation is allowed to vary in space. The tensorial nature of $\boldsymbol{a}$ is required to represent anisotropic materials, that deform preferentially in some directions (such as the ice mono-crystal, e.g. Meyssonnier and Philip, 1996; Gagliardini and Meyssonnier, 1999). In the case of an isotropic rheology, the fluidity $\boldsymbol{a}$ is a scalar $a$ times the unit tensor (e.g. Schulson and Duval, 2009). The constitutive material law given by Eq. (3) corresponds to viscoplastic creep, where the deformation occurs through dislocation movement within the ice material.

With the constitutive law Eq. (3), it can be shown (Tsuda et al., 2010) that the macroscopic strain rate of a sample under compression in a snowpack or in a firn column can be expressed in the form

$$\dot{E} = A(\mu,T)\Sigma^{n} \tag{4}$$

where $A$ is a scalar that does not depend on the overburden stress $\Sigma$, but depends on the snow/firn microstructure $\mu$ and also potentially on the temperature $T$. Physically, $A$ represents a form of fluidity, analogous to the fluidity tensor $\boldsymbol{a}$ at the microscopic scale. We also stress that the macroscopic non-linearity exponent $n$ in Eq. (4) is the same as in the microscopic deformation law of Eq. (3). In snow sciences, the viscous compaction of snow and firn is traditionally expressed in the form

$$\dot{E} = \frac{\Sigma^{n}}{\eta} \tag{5}$$

where we refer to $\eta = A^{-1}$ as the compactive viscosity (factor). Again, $\eta$ does not depend on the magnitude of the loading $\Sigma$, but only on the microstructure and the temperature of the sample. We note that the compactive viscosity is sometimes defined by the ratio $\Sigma/\dot{E}$ (e.g. Kojima, 1967; Wiese and Schneebeli, 2017) irrespective of the non-linearity of the constitutive law. While only the latter definition of the compactive viscosity has also physical units of a viscosity (i.e. Pa s), we do not follow



this convention here as in this case the compactive viscosity is not an intrinsic (microstructure and temperature dependent) property of the snow/firn sample. It would also depend on the overburden stress $\Sigma$.

Since the microscopic constitutive law characterizes visco-plastic processes in the ice, the microscopic rheology depends on temperature. This temperature-dependence is inherited by the macroscopic compactive viscosity $\eta$ (Kirchner et al., 2001).
Typically, the fluidity $\boldsymbol{a}$ in the microscopic law involves an Arrenhius factor with an activation energy $Q$, which implies the same temperature-dependence

$$A(\mu,T) = A_0(\mu)\exp\left[\frac{Q}{R}\left(\frac{1}{T_0} - \frac{1}{T}\right)\right] \tag{6}$$

at the macroscopic scale. Here, $A_0$ is a reference macroscopic fluidity that only depends on the microstructure, and $T_0$ a reference temperature. Re-expressed in terms of compactive viscosity this implies

$$\eta(\mu,T) = \eta_0(\mu)\exp\left[\frac{-Q}{R}\left(\frac{1}{T_0} - \frac{1}{T}\right)\right] \tag{7}$$

where $\eta_0 = 1/A_0$ is a reference compactive viscosity that only depends on the microstructure.

## 2.3   Viscosity ratios

For the comparison of the macroscopic mechanical behavior of different microstructures, it is helpful to utilize viscosity ratios. For two snow/firn samples under the same mechanical load $\Sigma$ at the same temperature, the ratio of their respective macroscopic
strain rates $\dot{E}_A$ and $\dot{E}_B$ is given by

$$\frac{\dot{E}_A}{\dot{E}_B} = \frac{\eta_B}{\eta_A} = \frac{\eta_{0B}}{\eta_{0A}} \tag{8}$$

The ratio is independent of any scalar prefactor in the microscopic deformation law. Indeed, halving the fluidity $\boldsymbol{a}$ results in a doubling of both $\eta_{0A}$ and $\eta_{0B}$, and thus their ratio remains unchanged. Viscosity ratios are an interesting metric to test the applicability of different ice rheologies to snow/firn modelling, as they remove the influence of a potentially poorly constrained
scalar prefactor in the microscopic deformation law, leaving only the non-linearity exponent $n$ and space dependence of the rheology as relevant parameters.

## 3   Data and Methods

### 3.1   Benchmark densification rates and $\mu$CT images

In order to compare our simulations with independent estimates, we selected different Alpine and Antarctic field campaigns
with available $\mu$CT data and complementary measurements that can be used to constrain observed densification rates. These estimates are used for the comparison.



### 3.1.1 Alpine snow

For Alpine snow, we rely on the RHOSSA campaign (Calonne et al., 2020). This extensive data set provides daily density profiles of a snowpack over the 2015-2016 snow season at the Weissfluhjoch observation site in the Swiss Alps. Four snow layers have been carefully tracked and measured with several instruments over the entire season, including a RG snow layer and a DH snow layer (following the classification of Fierz et al., 2009). From these data, the observed, macroscopic strain rate of a given layer can be estimated from

$$\dot{E} = \frac{\dot{\rho}}{\rho} \tag{9}$$

where $\rho$ and $\dot{\rho}$ are the density of a layer and its time derivative, respectively. By convention, the strain rate is positive in the case of compaction. The relevant overburden stress $\Sigma$ imposed on a given layer can be obtained by integrating the density profile above that layer to the snow surface

$$\Sigma = \int\limits_{\text{layer}}^{\text{surface}} dz\, g\rho(z) \tag{10}$$

where $\rho(z)$ is the density at height $z$ in the snowpack, and $g$ is the gravity acceleration.

The density time series of the layers were obtained on a daily basis using a snow penetrometer (SnowMicroPen; Calonne et al., 2020). The resulting data includes an apparent day-to-day variability that stems from spatial variability, resulting in a strongly fluctuating strain rate estimate when using Eq. (9) directly. This variability can cause an apparent decrease in density from one day to the other, which would be interpreted as a negative strain rate. To avoid such issues, the time evolution of the tracked snow layers was smoothed by visually selecting tie-points in the profiles, in order to reconstruct a piece-wise linear and strictly increasing density time series. In order to include the uncertainty of this method, a total of 25 time series were manually created for both the DH and RG layers. Their median values were taken as the observed strain rate of the layer and used to deduce the compactive viscosity. The upper and lower bounds of this procedure were used to characterize the spread of this method.

The density time series of the tracked layers in the RHOSSA campaign was regularly validated by $\mu$CT measurements, however at a much lower temporal resolution. Therefore the snow was regularly sampled and the microstructure was obtained with $\mu$CT at a resolution of $18\,\mu$m (Calonne et al., 2020). For our study, we selected $\mu$CT scans from the DH and RG layers, on the 13/01/2016 and 16/02/2016. This selection was motivated by the fact that on the 13/01/2016 the RG and DH samples have a similar density but different compaction rates, while on the 16/02/2016 the two snow layers have a similar compaction rate but different densities. Moreover, the DH and RG layer in the RHOSSA snowpack were almost adjacent, and therefore subject to a similar overburden stress and a similar temperature of about $-3\,^\circ C$. Differences between these two layers thus cannot be explained by a difference in stress or temperature, but should rather reflect a difference in their intrinsic mechanical properties.

These RHOSSA layers provide an ideal benchmark for simulations, which should be able to predict that DH snow tends to be much more resistant to compaction than RG snow (Kojima, 1967, 1975), an important feature of snow mechanics that models need to account for (Vionnet et al., 2012).



### 3.1.2 Antarctic firn

For the estimation of firn compaction rates with simultaneous microstructure measurements, we rely on an Antarctic field campaign. Our data originate from the B34 and B54 ice cores, both drilled on the East Antarctic plateau. The B34 core was drilled at Kohnen Station, a site characterized by a $10\,\mathrm{m}$ borehole temperature of $-44.5\,^{\circ}C$ (Weinhart et al., 2020), and the B54 was drilled near the OIR camp (displayed in Fig. 1 of Weinhart et al., 2020), a site with a measured $10\,\mathrm{m}$ borehole temperature of $-53\,^{\circ}C$.

For each firn core, a bulk density versus depth and an ice-age versus depth profiles were produced based on firn core weighting and temporal synchronization with other cores. Assuming that the firn density profile is in steady-state, the compaction rate of the firn column at a given depth can be estimated by

$$\dot{E} = \frac{1}{\rho}\frac{d\rho}{d\mathrm{Age}}\,. \tag{11}$$

Here $\frac{d\rho}{d\mathrm{Age}}$ is the derivative of the density profile with respect to the age profile. To obtain the density versus age profile, we
combine the density-depth and the age-depth profiles. Similarly to the case of the Alpine snowpack, the overburden stress $\Sigma$ for a layer at a given depth is calculated from the integration of the density of the overlying firn column.

As the density profiles were obtained by weighting $1\,\mathrm{m}$ long cores, they do not resolve layer-to-layer variability and the derived strain rate can already be regarded as an averaged (smoothed) bulk value. The strain rate of an individual layer in $1\,\mathrm{m}$ core might still differ from this bulk value if its mechanical and microstructural properties differ from the average ones in the
core.

While the firn density profiles are therefore sufficiently smooth and do not include any decrease of density with depth (which would be interpreted as expansion in Eq. 11), the deduced strain-rate profiles are still fluctuating, resulting in large and non-physical variations of the compactive viscosity at the meter scale. Consistent with the Alpine case, we therefore smoothed the profiles by manually selecting tie-points to create a piece-wise linear profile from which the compactive viscosity is deduced.
In order to characterize the uncertainty of the method, profiles corresponding to the outer envelops of the strain rate profile were also created and used to derive upper and lower bonds for the strain-rate profiles.

Finally, these firn core data are complemented by 36 $\mu$CT scans (4 from B34 with a $40\,\mu\mathrm{m}$ resolution, 32 from B54 with a $30\,\mu\mathrm{m}$ resolution), with ice volume fractions ranging from $0.428$ to $0.933$ (that is to say densities ranging from about 390 to $860\,\mathrm{kg\,m^{-3}}$, assuming a density of ice of $917\,\mathrm{kg\,m^{-3}}$).

## 3.2 Finite Element Modeling

The experimental observations of snow and firn compaction were complemented with simulations using the finite element method (FEM). The goal of these simulations is to estimate the compaction rate of a sample based on its microstructure and a given microscopic ice rheology. Our simulation workflow, from $\mu$CT scanning to the estimation of the compactive viscosity, is schematically summarized in Fig. 1 and detailed below.



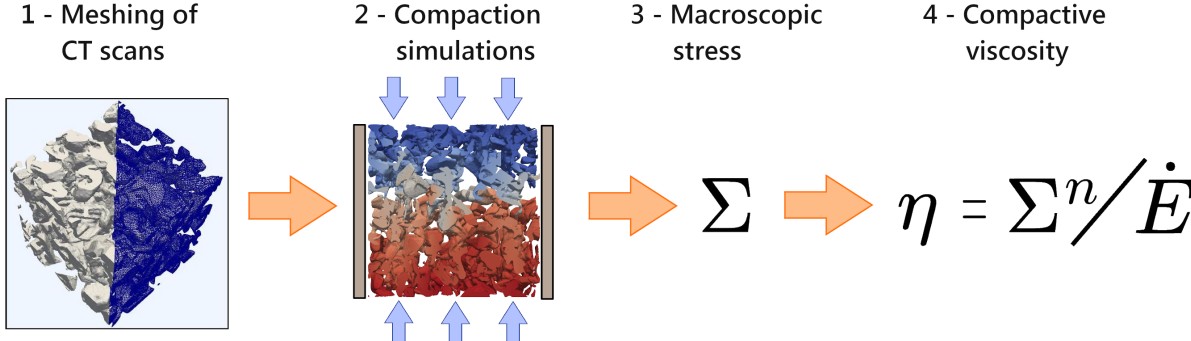

**Figure 1.** Simulation workflow used in this study, in order to estimate the compactive viscosity of a snow/firn sample: 1- Production of a tetrahedral mesh from the $\mu$CT images using CGAL, 2- Simulation of strain rate-imposed compaction using ElmerFEM (the color-scale stands for the vertical velocity), 3- Computation of the resulting macroscopic stress exerted at the top and bottom of the sample, and 4- Computation of the compactive viscosity of the microstructure.

### 3.2.1 Image segmentation and mesh generation

The first step of our simulation workflow is to produce tetrahedral meshes representing the snow/firn microstructures. As detailed below, we performed simulations using both isotropic and anisotropic constitutive laws for the ice. Depending on whether a simulation is performed with an isotropic or an anisotropic material, the mesh generation process was slightly different.

For the isotropic ice simulations, each $\mu$CT image was first segmented into a binary voxel image of ice and air (as detailed in Calonne et al., 2020) which was used as input for the CGAL[1] meshing library (Fabri et al., 2000) in order to produce a tetrahedral mesh of the ice matrix. Disconnected parts of the mesh were removed by component labeling, in order to obtain a simply connected region for the ice microstructure. The last step is necessary as disconnected regions cannot carry mechanical loads and lead to an ill-defined mathematical problem in the finite element formulation.

As the goal of using an anisotropic ice material was to model snow microstructure as an ensemble of mono-crystals with a crystallographic texture, the $\mu$CT images were also segmented into individual ice crystals as in Theile and Schneebeli (2011), Hagenmuller et al. (2014), or Willibald et al. (2020). As $\mu$CT does not carry any information about the crystallographic orientation of the ice, this segmentation was done on a purely geometrical basis using a watersheding algorithm, following Willibald et al. (2020). The resulting segmented images for two RHOSSA samples are shown in the left column of Fig. 2. The segmentation appears to be realistic when compared to c-axis orientation measurements in snow from thin-sections (Riche et al., 2013, Fig. 3). The crystal-segmented $\mu$CT images were then meshed using the CGAL software and disconnected parts removed, following the same procedure as for the standard binarized $\mu$CT images. Visualisations of segmented 3D FEM meshes, composed of various individual ice crystals, are displayed in the right column of Fig. 2.

---

[1]https://www.cgal.org/





**Table 1.** Description of the alpine snow and Antarctic firn $\mu$-CT images used for finite elements simulations.

https://www.overleaf.com/project/64537e027d0b3cca12c87549

| Sample(s) | Voxel size ($\mu$m) | Sample(s) size (cm$^3$) | Ice Volume Fraction | Temperature ($^\circ$C) |
|---|---|---|---|---|
| RHOSSA RG 13/01/2016 | 18 | $1.08 \times 1.08 \times 1.08$ | 0.27 | -3 |
| RHOSSA DH 13/01/2016 | 18 | $1.44 \times 1.44 \times 1.44$ | 0.28 | -3 |
| RHOSSA RG 16/02/2016 | 18 | $1.44 \times 1.44 \times 1.08$ | 0.46 | -3 |
| RHOSSA DH 16/02/2016 | 18 | $1.44 \times 1.44 \times 1.44$ | 0.30 | -3 |
| B34 | 40 | $1.2 \times 1.2 \times 1.2$ | 0.43 to 0.93 | -44.5 |
| B54 | 30 | $1.8 \times 1.8 \times 1.8$ | 0.60 to 0.80 | -53 |

In each case, we used the full size of the $\mu$CT images in order to have as representative as possible volumes. Specifically,
the size of the RHOSSA snow samples corresponds to 1.44x1.44x1.44 cm$^3$ cubes for both the DH samples and the RG sample of the 16/02/2016, and a 1.08x1.08x1.08 cm$^3$ cube for the RG sample of the 13/01/2016. The B34 samples correspond to 1.2x1.2x1.2 cm$^3$ cubes and the B54 samples to 1.8x1.8x1.8 cm$^3$ cubes. The samples used for FEM modeling and their characteristics are summarized in Table 1.

### 3.2.2   Finite element solution

Once a tetrahedral mesh is obtained, the Stokes equations Eq. (1) are solved using the finite element method to estimate the compactive viscosity of a given microstructure for a given microscopic rheology as outlined in Section 2. The simulations were carried out with the ElmerFEM[2] software, regularly used in ice sheets modeling. In this way, we can build on existing work in the context of ice flow that is typically done on larger scales (Gagliardini et al., 2013; Law et al., 2023). As our meshes
typically contain ten millions of elements, we used the ParStokes equation solver, that was specifically developed for solving the Stokes equations for a large number of elements on a parallel computer (used for instance in  Schannwell et al., 2020). However, the ParStokes solver was originally developed for isotropic materials only. For the purpose of our study, we extended the ParStokes solver to cope also with a linear transverse isotropic material (see Appendix A), in order to emulate the behavior of ice monocrystals (Gagliardini and Meyssonnier, 1999; Burr et al., 2017). With this transverse isotropic rheology, shearing
in the basal plane (the plane perpendicular to the c-axis of the crystal) is set to be $100$ times faster than shearing perpendicular to it (Burr et al., 2017).
For solving the linearized FEM equations, we relied on the BiCGSTAB or the GMRES iterative methods, and convergence was assumed when the relative residual of the system reached at least $5 \times 10^{-5}$ (some simulations reached a lower convergence criterion, but others showed quite slow convergence rates after passing $1 \times 10^{-4}$ which prevented reaching smaller criteria).

---

[2]http://www.csc.fi/elmer



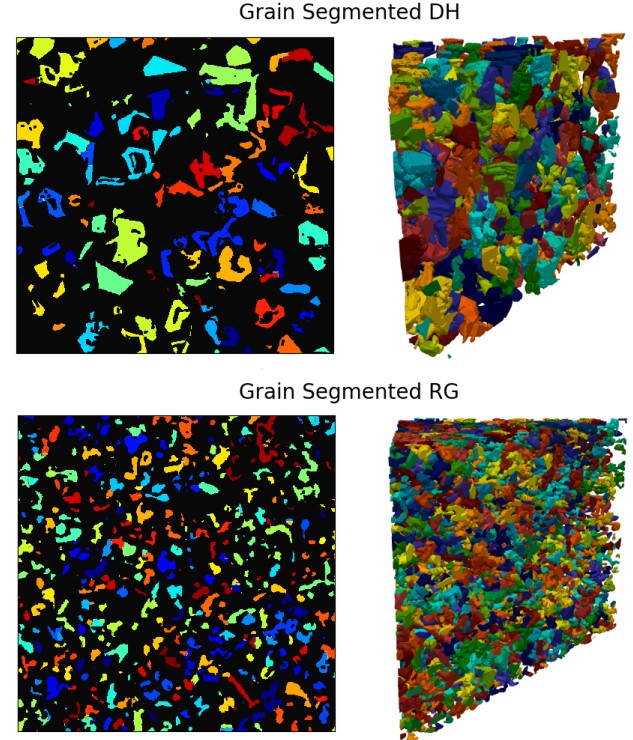

**Figure 2.** Geometrical segmentation of $\mu$CT images into individual grains (left) and the corresponding 3D meshes (right). Each grain is represented by a given color. Samples correspond to the DH and RG snow sampled on the day 13/01/2016 of the RHOSSA campaign.

For non-linear problems, the non-linear iterations were considered to have converged when the relative difference between two consecutive iterations was smaller than $1 \times 10^{-4}$.

### 3.2.3 Boundary conditions

For the simulations, boundary conditions need to be prescribed. For the compression of snow and firn samples, we used the so-called Periodicity compatible Mixed Uniform Boundary Conditions (PMUBC; Pahr and Zysset, 2008). The vertical strain

rate is imposed by prescribing the top and bottom vertical velocities, while a vanishing normal velocity is imposed on the sides. The advantage of such boundary conditions is twofold. First, these conditions mimic the natural, laterally constrained situation during compaction in snowpacks and firn columns. Second, these boundary conditions require the smallest volumes to achieve a representative behavior (Pahr and Zysset, 2008).

Once the simulation has completed, the total reaction forces acting on the top and bottom faces are calculated. The macro-

scopic overburden stress is computed as the resulting average force divided by the sample surface area. Finally, the compactive viscosity $\eta$ of the sample is obtained as the ratio between the macroscopic stress, with the appropriate non-linearity exponent, and the prescribed macroscopic strain rate, as given by Eq. (5).



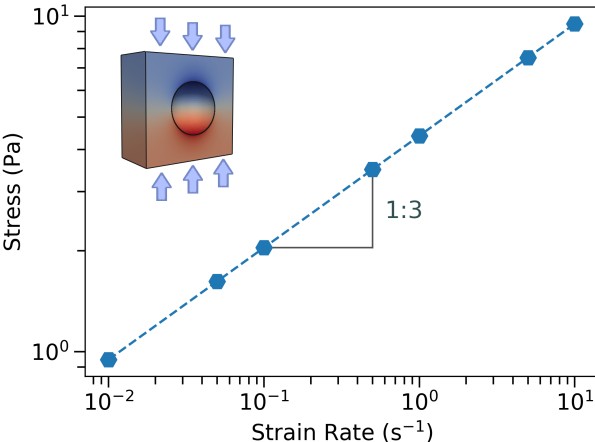

**Figure 3.** Stress versus Strain rate curve characterizing the response of a microstructure composed of isotropic ice under compression. The macroscopic response follows a power-law with the same exponent as the ice material, here with $n = 3$. A cut-view of the microstructure, a cube with a hollow sphere inside, is displayed in the top left corner. The blue-to-red color scale indicates the vertical velocity field.

### 3.3 Testing the finite element setup

To provide some confidence in the correctness of the finite element setup, we conducted two numerical test experiments.

First, we verified that the macroscopic response of a microstructure with the microscopic constitutive law (3) follows the same power-law with the same stress-exponent $n$. To this end we used a simple, spherical inclusion microstructure consisting of a cube with a hollow sphere in its center (displayed in the top left corner of Fig. 3). The deformation law chosen for the constituting material is an isotropic power-law with $n = 3$ and a fluidity pre-factor $a = 1\mathrm{Pa}^{-3}\,\mathrm{s}^{-3}$. Using our finite element framework, the microstructure was deformed with different imposed macroscopic strain rates, and the macroscopic stress was

obtained from the output of the simulations. The results are shown in Fig. 3 and confirm that the macroscopic strain rate versus stress curve indeed follows a power-law with $n = 3$, as predicted by Eq. 5. The corresponding compactive viscosity of this simple microstructure can be evaluated to $\eta = 85.1\,\mathrm{Pa}^3\,\mathrm{s}$ (and this value of course depends on the specific choice of the non-linear exponent $n$ and of the fluidity pre-factor $a$).

Second, we tested the anisotropic behavior of the implemented transverse isotropic constitutive law which is supposed to

represent the rheology of an ice mono-crystal. To this end, we performed simulations of the compression of cylinder composed of a single mono-crystal with flatten top and bottom which is referred to as the flattened Brazilian test (Wu et al., 2018). For the boundary conditions, the bottom boundary of the sample is fixed while a constant vertical velocity is imposed at the top. The results of the simulations are displayed in Fig. 4, and show the appearance of the well-known shear band when the crystallographic orientation is tilted compared to the direction of compression. While the precise position and orientation of

this shear band are non-trivial and depend on the geometry of the sample (notably through stress concentrations developing near corners, as visible in e.g. Fig. 8 of Patel and Martin, 2018), such a shear band is characteristic for an anisotropic material





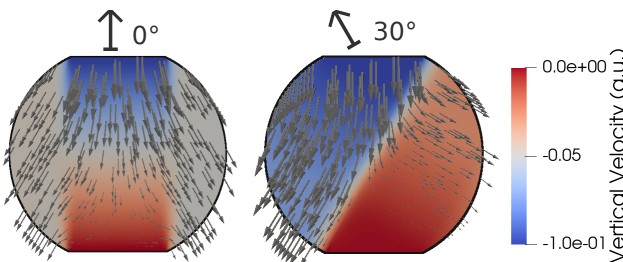

**Figure 4.** Compressive response of a flattened cylinder composed of a transverse isotropic material, depending on the orientation of the crystal axis ($0°$ and $30°$ compared to the vertical). The small gray arrows indicate the velocity of the ice material, and the color-scale indicates the vertical component of the velocity field.

and does not appear in the isotropic case. Moreover, the resulting macroscopic stress required to obtain the same imposed macroscopic strain rate is about 23 times smaller in the case of a tilted c-axis, showing the softening of the material depending on the respective orientation between the compression and the crystal.

## 4   Results and Discussion

We start with the simplest rheology for the ice matrix commonly used in literature, namely Glen's law for isotropic, polycrystalline ice.

### 4.1   Firn is a foam of polycrystalline ice - snow is not.

Several works in literature proposed that the deformation of the ice matrix in snow and firn is similar to the deformation of isotropic polycrystalline ice, i.e. glacier ice. This idea dates back to Mellor and Smith (1966), where the deformation of snow was experimentally studied alongside polycrystalline ice to unravel the similarities between the two. This approach has been supported by subsequent work of Kirchner et al. (2001), who concluded that the viscous compaction of snow has the same non-linear properties as polycrystalline ice. In this respect, snow is viewed as a "foam of ice" (Kirchner et al., 2001), or more precisely a foam of polycrystalline ice. The viscoplastic deformation of polycrystalline ice is nowadays reasonably well understood and usually cast into Glen's law for an isotropic power-law rheology with $n = 3$ and tabulated fluidity values depending on the temperature of the ice (Schulson and Duval, 2009; Cuffey and Paterson, 2010). As the benchmark densification rates for snow and firn were obtained with temperatures around $-3$, $-44.5$, and $-53\,°\mathrm{C}$, the corresponding ice fluidity are as $1.7 \times 10^{-24}$, $5.2 \times 10^{-27}$, and $1.4 \times 10^{-27}\,\mathrm{Pa}^{-3}\,\mathrm{s}^{-1}$ respectively.

Our first attempt was thus to simulate the deformation of snow and firn assuming Glen's law as the rheology for the ice matrix. Fig. 5 shows the comparison between the compactive viscosities obtained with FEM simulations and those derived from the experimental data.



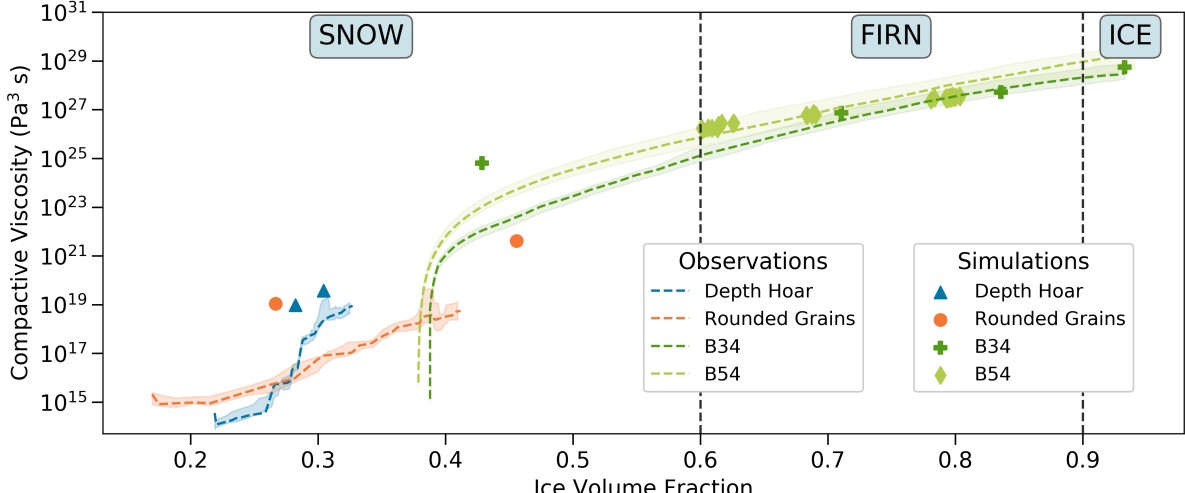

**Figure 5.** Comparison between simulated and observed compactive viscosities of snow and firn samples, assuming Glen's law for the ice rheology. The Depth Hoar and Rounded Grains samples and data are taken from the RHOSSA campaign, and the B34 and B54 ones from the firn cores. The ice volume fractions of the simulated points are computed based on the CT scans. The shaded areas correspond to the uncertainties in deriving the compactive viscosity values, as described in Section 3.1

.

Concerning firn (ice volume fraction above 0.6), the Figure shows a general agreement, despite a cluster of simulated B54 samples that appear not viscous enough. Our interpretation is that since all these samples were taken at a similar location in the firn, this very location in the firn column may not be representative for the bulk and steady-state firn column due to the vertical variability existing in firn (Hörhold et al., 2011; Fourteau et al., 2019). Concerning snow (ice volume fraction below 0.6), Fig. 5 reveals a large overestimation of the compactive viscosities for all samples. This overestimation of the compactive viscosity when using Glen's law is consistent with the results of Theile et al. (2011), that reached a similar conclusion. The overestimation also exceeds the uncertainties in the observations that were computed as explained in Sec. 3 and shown as shaded areas in Fig. 5. Moreover, while the RHOSSA observations indicate that on the 13/01/2016 and 16/02/2016 the DH sample is, respectively, about 5 times and 1 times as viscous as the RG sample, our simulations using Glen's law predict a DH sample that is, respectively, 0.83 and 0.0093 times as viscous as the RG sample. Thus, not only is Glen's rheology largely overestimating the viscosity of snow, it also fails in explaining relative differences (viscosity ratios) and as to why DH is such a viscous snow type compared to RG at the same density, temperature, and overburden stress.

Our results confirm that Glen's law is appropriate to model firn compaction, but clearly another microscopic ice rheology is required to explain the viscous compaction of snow.



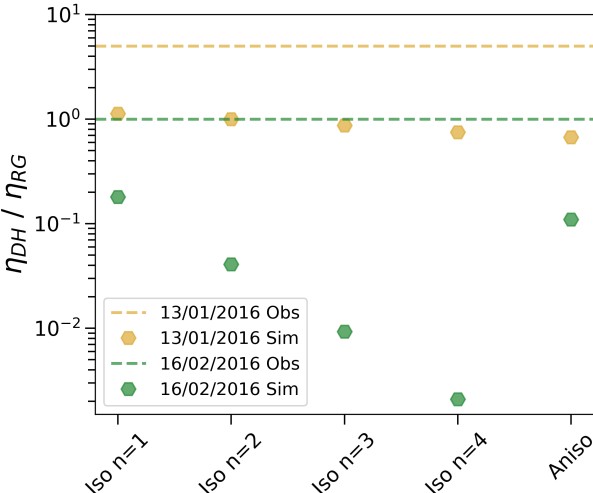

**Figure 6.** Comparison of simulated compactive viscosity ratios (DH over RG, markers) with observations (dashed lines) from 13/01/2016 and 16/02/2016 for different versions of the micro-scale constitutive law (x-axis).

## 4.2 Sensitivity study of isotropic rheologies

The disagreement of Glen's (isotropic polycrystalline) law could be resolved in a pragmatic but physically unjustified way, by artificially increasing the fluidity $a$ in Glen's law. Such a rheology could still be an interesting trade-off between physical rigor and practical considerations, as long as the obtained simulated compactive viscosities reasonably reproduce experimental observations. For instance, by using an ice rheology with adjusted parameters, different from Glen's law, Wautier et al. (2017) was able to simulate compaction rates that are much more in line with experimental observations. However, such a fudge-factor would still be in contradiction with our results: As explained in Section 2, the viscosity ratio between samples is independent of any potential (unjustified) modification of the ice fluidity $a$. In other words, while modifying the fluidity of the ice can be used to attenuate the general overestimation of the simulated snow viscosity, it would still fail in explaining why the simulated compactive viscosity of rounded grains is (relatively) too high when compared to DH snow Fig. 5. Since this relative difference between RG and DH is a prominent feature in the densification of snow, we conclude that isotropic power-law rheologies with $n = 3$ are not suited for snow modeling.

Detailed snow models rather employ a Newtonian rheology ($n = 1$) instead (Bartelt and Lehning, 2002; Vionnet et al., 2012). Macroscopic Newtonian rheologies are also used in low-density firn studies (Schultz et al., 2022). Though employing the same rheology on the micro-scale in our simulations cannot be directly motivated by an underlying physical argument, related numerical experiments will comprehend our understanding of homogenization in snow compaction. Accordingly, we conducted a sensitivity analysis on microscopic, isotropic constitutive laws using different exponents ($n = 1$, $n = 2$, and $n = 4$) as similarly done by Wautier et al. (2017). For these simulations, physically-constrained values for the scalar fluidity $a$ of the ice rheologies are not available. Hence, comparing simulated and observed *absolute* values of compactive viscosities is





not meaningful. We therefore focus here on viscosity ratios of DH and RG during the RHOSSA campaign, as this ratio is independent of the ice fluidity.

The results of our sensitivity study are shown in Fig. 6 and summarized in Table 2. Similar to Glen's law for both investigated days, and independent of the microscopic rheology, the simulations predict a viscosity ratio that is not in agreement with the observations. One may either consider this as RG snow being too viscous or DH snow being not viscous enough. Moreover,

increasing the non-linearity exponent $n$ leads to a decrease in the viscosity ratio moving farther away from the observations. We interpret this result through the differences in geometry between RG and DH. DH is characterized by smaller necks than RG. This leads to higher local stress concentrations in DH than in RG, which can be confirmed from the stress distribution in the DH and RG samples. These higher stress concentrations result in higher local strain rates, which are exacerbated by the non-linear nature of the power-law rheology (for $n > 1$). This effect is particularly marked on the DH and RG samples from

the 16/02/2016, as shown by the sharp decrease of the DH viscosity compared to that of RG as $n$ increases. Thus, further increasing the value of $n$ beyond $n = 4$ appears unlikely to explain the compactive behavior of DH and RG snow. The insight from this sensitivty study consolidates our conclusion that the viscous compaction of snow cannot be explained by assuming that the ice matrix deforms according to a simple isotropic power-law, might it be linear or non-linear.

### 4.3 Snow as an ensemble of monocrystals

The inability of Glen's (isotropic polycrystalline) law to explain snow viscous compaction is consistent with the fact that microstructural geometrical grains in snow are in fact mono-crystalline units (Riche et al., 2013). The mechanical behavior of polycrystalline ice arises from the collective behavior of neighboring mono-crystals blocking each other as their preferential directions of deformation are not compatible with one another. In contrast, the monocrystalline grains in snow have many free surfaces at the ice/pore interfaces, where the crystals are free to deform (Scapozza and Bartelt, 2003). Thus, the ice

matrix composing snow can be expected to deform much more easily than polycrystalline ice. Note that this reasoning is also compatible with the applicability of Glen's law in firn. Here, due to the lower porosity, there are not so many interfaces with the pores, and individual ice-crystals tend to block one another, resulting in an ice rheology close to that of polycrystalline ice. Such a transition in behavior, which is driven by as a transition in density, is widely adopted (Alley, 1987; Arnaud et al., 2000; Salamatin et al., 2009; Morris et al., 2022)

Therefore, as a natural generalization of our work on snow compaction, we extended our study to an anisotropic rheology, in order to advance in the direction of representing the grains in snow as monocrystals. To this end, we conducted simulations using a transverse isotropic rheology (Meyssonnier and Philip, 1996; Gagliardini and Meyssonnier, 1999). In contrast to the isotropic rheologies used above, this anisotropic rheology introduces a novel feature at the micro-scale, namely anisotropic deformations of grains, that shear much more easily in their basal plane than in other directions (Montagnat et al., 2014b).

This idea of modelling snow as an ensemble of monocrystals has been previously proposed by Theile et al. (2011). They found that this rheology yielded smaller compactive viscosities, more in line with observed values. However, Theile et al. (2011) considered only RG microstructures on geometrically simplified meshes. Therefore, the relative comparison between DH and





**Table 2.** Ratio between the DH and RG compactive viscosities from various ice rheologies.

| Day | Isotropic $n=1$ | Isotropic $n=2$ | Isotropic $n=3$ | Isotropic $n=4$ | Anisotropic | Observations |
|---|---|---|---|---|---|---|
| 13/01/2016 | 1.13 | 1.0 | 0.87 | 0.75 | 0.67 | $\sim 5$ |
| 16/02/2016 | 0.18 | 0.041 | 0.0093 | 0.0021 | 0.11 | $\sim 1$ |

RG in terms of viscosity ratios using the full microstructure constitutes an important cross-validation of this finding related to the aim of the present work.

For the simulation, the ice crystals were randomly binned into 100 different crystallographic orientation classes, each with a given random c-axis orientation. The distribution of the c-axes orientation corresponds to an isotropic texture (i.e. with no preferential orientation for the c-axes). In order to simplify the comparison between the DH and RG snow simulations, the same c-axes distribution was used for snow samples taken from the same day. For the present purpose, we limited ourselves to the *linear* transverse isotropic rheology that has been proposed for ice (as in Gagliardini and Meyssonnier, 1999).

Results of the simulations for the anisotropic rheology are shown in Fig. 6 and Table 2. They confirm the same disagreement that was found with the simulations for the isotropic rheology: the viscosity ratio between DH and RG is too low. Comparing the viscosity ratios from the anisotropic model with their isotropic counterpart with the same exponent ($n=1$) shows that the performance is even reduced, with a slight decrease of the viscosity of DH relative to RG. The use of a linear transverse isotropic rheology, meant to represent monocrystal behavior, cannot be considered more realistic. While we have only considered the

linear anisotropic case here, it can be expected that a non-linear anisotropic material law (when using exactly the same grain segmentation) would further increase the difference to the observations, for the same reasons as detailed in the isotropic case.

To illustrate the main differences between the anisotropic and the isotropic constitutive law, we computed the deviatoric stress field in the RHOSSA 13/01/2016 RG microstructure for both material laws. The distribution of the $s_{33}$ component (i.e. vertical component) of the deviatoric tensor is displayed in Fig. 7 a. It reveals that overall the two deviatoric stress distributions are relatively similar, once normalized by the macroscopic stress. Only the tails of the distribution differ. Thus, while stress

concentrations could be expected due to deformation incompatibilities of neighbouring crystals, they appear to be relatively limited. This is confirmed by Fig. 7 b, which displays the spatial field of the normalized $s_{33}$ component within a slice of the RG microstructure. The anisotropic and isotropic cases show similar spatial patterns of stress concentrations, which appear to be dictated by ice matrix geometry rather than crystallographic effects. In our simulations, the stress pattern driving the

deformation is only little affected by the use of an anisotropic material with crystallographic orientations. In the case of an isotropic crystallographic texture, where crystal orientations are not correlated with zones of stress concentration, the RG and DH can be expected to be impacted in a similar fashion by their random crystal orientation in zones of stress concentrations. Thus, the use of an anisotropic material with planes of preferential deformation modifies the compactive viscosity of a given sample compared to the isotropic case, but the viscosity ratio between two samples remains relatively constant (cf. Fig. 6).






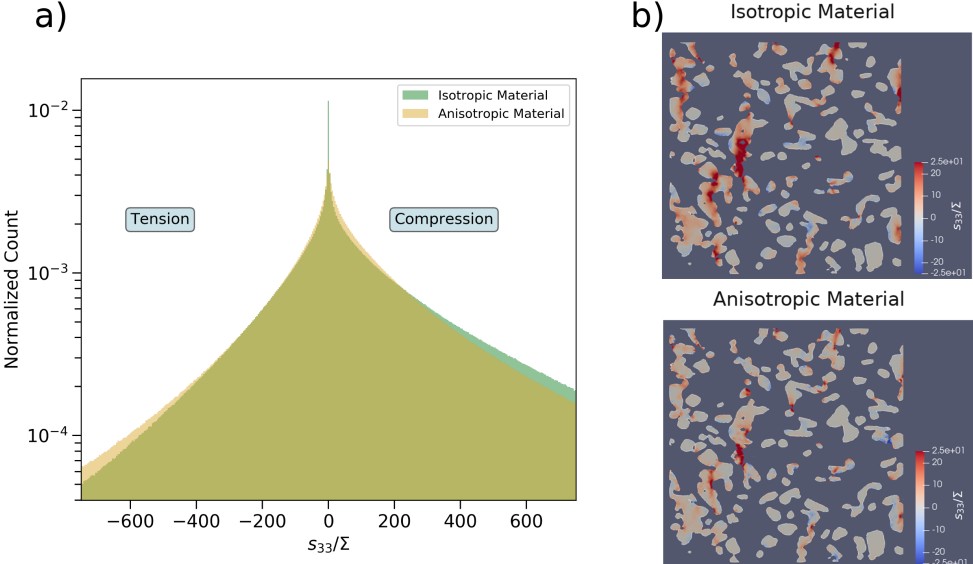

**Figure 7.** a): Distribution of the $s_{33}$ component of the deviatoric tensor normalized by the macroscopic stress $\Sigma$ in an RG microstructure, using either an isotropic or anisotropic constitutive law for the ice matrix. b): Field of the normalized $s_{33}$ component of the deviatoric tensor in a RG microstructure, using either an isotropic or anisotropic constitutive law for the ice matrix. Red colors stand for compressive zones and blue colors for tensile zones.

## 4.4 Perspectives for the viscous compaction of snow and firn

By considering the viscous compaction of snow and firn in the same computational, microstructure-based framework, we were able to support and contradict underlying mechanisms that were previously hypothesized in the snow and firn literature.

The overall agreement of the isotropic simulations with observed firn densification rates using a published fluidity for the actual firn temperature, supports the notion of firn as a foam of (isotropic, polycrystalline) ice (Kirchner et al., 2001) where the deformation stems from (intra-crystalline) dislocation creep(Schulson and Duval, 2009). We note that in our study, the rheology for polycrystalline ice was taken from Cuffey and Paterson (2010), which reports a non-linear exponent $n = 3$. However, the recent experimental study of Li and Baker (2022) on firn compaction rather reports a non-linear exponent of $n \sim 4$. As noted by Li and Baker (2022), a non-linear exponent closer to $4$ is also compatible with observations of polycrystalline ice (e.g. Goldsby

and Kohlstedt, 2001; Bons et al., 2018). While the question of the best choice for $n$ remains open, these inconsistencies in the specific $n$ value do not contradict our result that firn compaction can be adequately simulated using an ice rheology adjusted on polycrystalline ice observations. Remaining differences between simulations and firn core observations could be further explored if microstructure profiles were continuously available at high vertical resolution, similar to (Montagnat et al., 2020). Thereby, the separation of scales between observations (strain rates averaged over one meter) and simulations (cm-sized

samples) and the associated uncertainty could be further narrowed down.



In contrast, the notion of snow as a foam of ice following the same creep mechanism as high density firn is clearly ruled out by our study. While this result seems trivial in view of the number of studies highlighting the difference between snow and firn (e.g. Herron and Langway, 1980; Arnaud et al., 2000; Morris et al., 2022) we stress that these differences were hitherto never explored by microstructure-based simulations.

Our results for snow re-initiate the question about the dominant deformation mechanisms. By using viscosity ratios (cf. Fig.6), we were able to show that DH and RG snow cannot be consistently modelled by using the same micro-scale constitutive law (isotropic or anisotropic), strongly suggesting a fundamental difference in the underlying physics. Viscosity ratios provide a complementary benchmark, since biases resulting from the numerics or the prefactor in the micro-scale constitutive law should cancel out.

On one hand, this raises the question of whether the agreement for the anisotropic rheology found in Theile et al. (2011) would still hold when including different snow types at the same temperature and relaxing the mesh simplification step in representing the microstructure. Likewise, exploring anisotropic rheologies should be extended further, for instance by including non-linearity effects (Schulson and Duval, 2009; Montagnat et al., 2014b) or a preferential orientation of the c-axes (Riche et al., 2013). Our extension of the ParStokes solver in ElmerFEM could provide an efficient modelling starting point in the
direction of transverse-isotropic, viscous rheologies with a texture. Concurrent measurements of the texture are then unavoidable. Alternatively, micro-scale constitutive laws may be directly adopted from crystal plasticity, and simulated using dedicated numerical techniques based on Fast Fourier Transformation (Knezevic et al., 2009; Hure, 2019).

On the other hand, our viscosity ratios also raise the question of whether the density (of snow or firn) is actually the relevant property that discerns between different deformation mechanisms (Alley, 1987; Morris et al., 2022). Our snow samples were
selected to include cases (Fig. 5) with virtually the same density but clear differences in the observed densification rates. While the comparison to observations (for snow and firn) is subject to the same uncertainty (separation of scales between cm-sized simulations and layer-averaged densification rates in the observations) care needs to be taken in assuming a density-driven transition to a different deformation mechanism, e.g. grain boundary sliding (GBS) (Raj and Ashby, 1971; Langdon, 2006). Such a GBS mechanism provides an alternative to the intra-crystalline deformation discussed so far (Theile et al.,
2011). Indeed, it is often assumed that at low density the deformation of snow occurs through localized stress relaxation at the junctions between grains (Alley, 1987; Salamatin et al., 2009; Schultz et al., 2022). However, the implementation of such a deformation mechanism in FEM, and an extension of the sensitivity analysis Fig. 6 to fundamentally different forms of the micro-scale constitutive law Eq. 3, is not straightforward. Standard FEM techniques are not suited for localized, discontinuous deformations. Simulations of grain boundary sliding at the micro-scale would require more complex numerical methods, such
as the Extended Finite Elements Method (Khoei, 2014). This method has been developed to account for discontinuities and has been successfully applied to model grain boundary sliding in the past (Simone et al., 2006).

Finding the relevant drivers for transitions in the compactive viscosity is even complicated by recent experimental studies such as Wiese and Schneebeli (2017). This work found an immediate increase of the compactive viscosity during temperature gradient metamorphism (compared to isothermal samples), despite the absence of strong differences in structural parameters
such as density, SSA, or structural anisotropy. We believe that this kind of studies constitute a good experimental direction





to identify active, microscopic deformation mechanism(s), and should be able to explain why very small microstructural differences can lead to large compactive viscosity differences. Moreover, the acquisition of $\mu$CT images during the controlled deformation of snow in the laboratory could help to identify the microscopic mechanism(s) at play during deformation and be a guide to select the appropriate physics for microstructure-based simulations.

Besides snow compaction, the simulations using a transverse isotropic rheology showcased that the ability of ElmerFEM's ParStokes solver to handle highly-parallelized simulations (as in Schannwell et al., 2020) can be extended to account for complex anisotropic rheologies. Such a possibility could for instance be useful for large-scale ice sheet modeling, where the ice material can present an anisotropic behavior due to texture development (e.g. Gillet-Chaulet et al., 2006; Montagnat et al., 2014a). As mentioned in the Appendix A, the ParStokes solver relies on an approximation of a Schur-complement for block-
preconditioning. A path of future development for the ParStokes solver, and its ability to robustly handle anisotropic materials, could be to derive a better approximation of the Schur-complement, which could improve the block-preconditioning stage.

## 5    Conclusions

Modelling the viscous densification of snow and firn directly from the microstructure of samples constitutes an important step towards replacing empirical parametrizations in models with physics-based laws. These computationally demanding,
microstructure-based simulations can now be conveniently carried out with the required accuracy (mesh representation) using parallel computing. A holistic snow and firn densification picture is still hampered though by the limited insight into the micro-scale rheology of the ice matrix. This study explored several rheologies in microstructures taken from field campaigns and compared them to independent estimates. Using firn cores drilled in East Antarctica, our simulations largely confirmed that the ice matrix deforms according to an isotropic polycrystalline rheology, as classically used to model glacier or ice sheet
ice. For snow, none of the tested rheologies (isotropic vs anisotropic, linear vs non-linear) is able to quantitatively predict the large viscosity ratio between depth hoar and rounded grains, which is a critical requirement in snowpack modelling. Future (experimental and numerical) work is urgently needed to further constrain the form and the parameters in the micro-scale constitutive law of ice in snow.

## Appendix A:    Implementation of an anisotropic rheology in ElmerFEM

For the implementation in ElmerFEM (or any other FEM framework) of an anisotropic ice rheology, one needs to define the constitutive law that relates the deviatoric stress tensor $s$ to the strain rate tensor $\dot{\epsilon}$. In the case of a linear and anisotropic rheology, this constitutive law can be applied thanks to a fourth-order viscosity tensor $M$, i.e. $s = M : \dot{\epsilon}$. While we do not consider this case in our study, one could also use a non-linear rheology. For this purpose, this constitutive law would need to be linearized around the current estimate of the solution, and this linearized law expressed through a fourth-order tensor. The
solution would then be iteratively approached (typically through Picard or Newton iterations).



To represent the deformation of a monocrystal, we used the constitutive law of Gagliardini and Meyssonnier (1999), given in Voigt notation in Eq. 7 of their article. This constitutive law represents a transverse isotropic ice material with its c-axis oriented towards the vertical. The viscosity while shearing in the plane perpendicular to this c-axis is assumed to only be a fraction of the viscosity while shearing in the planes containing the c-axis. In our case, we set this fraction to be $0.01$ (following Burr et al., 2017). Note that in Gagliardini and Meyssonnier (1999) the constitutive law is expressed as the relation yielding $\dot{\epsilon}$ as a function of $s$, i.e. $\dot{\epsilon} = a : s$, with $a$ a fluidity tensor. For the implementation in ElmerFEM, such law needs to be inverted in order to have $s = M : \dot{\epsilon}$. Due to the incompressibility of ice, the computation of $M$ from $a$ is not unique (Loredo and Klöcker, 1997). However, as the Voigt representation of the $a$ tensor in Gagliardini and Meyssonnier (1999) is invertible, we can simply take the inverse as the tensor $M$.

The tensor $M$ obtained following the expression of Gagliardini and Meyssonnier (1999) is only valid for an ice crystal having its c-axis orientated vertically. In order to be applied in a 3D FEM simulation, this constitutive law needs to be rotated according to the actual orientation of the crystal. This is achieved through rotations matrices and using the co-latitude and longitude angles of the c-axes (Meyssonnier and Philip, 1996; Gagliardini and Meyssonnier, 1999).

Finally, one of the particularities of the ParStokes Solver used in this work is its use of block-preconditioning. Ideally, this block-preconditioner would be based on a Schur-complement (similarly to e.g. Worthen et al., 2014). However, as the computation of a Schur-complement can be costly, the isotropic version of the ParStokes solver approximates it as a FEM mass matrix divided by the element-wise scalar viscosity. However, in the case of anisotropy, the viscosity is not a scalar anymore. For the approximation of the Schur-complement in the anisotropic version of ParStokes, we instead used the first invariant of the viscosity tensor (Betten, 1982), normalized such that it equals the scalar viscosity in the limiting case of an isotropic rheology.

*Code and data availability.* The code used to run the FEM simulations will be provided upon direct request to the corresponding author. The RHOSSA data are available at https://www.envidat.ch/dataset/wfj_rhossa (doi:10.16904/envidat.151).

*Author contributions.* The research subject was designed by HL with inputs from KF. HL received the funding. The research was performed by KF with the help of HL. MM contributed to the implementation of the FEM Solver used in this work, and its extension to anisotropic materials. Firn core data were obtained by JF. The manuscript was written by KF and HL with inputs from JF and MM.

*Competing interests.* The authors declare having no competing interests.

*Acknowledgements.* This work was funded through the WSL Innovative Project call (grant number 202011N2133) and Swiss National Science Foundation (grant number 200020_178831). KF current position is funded by the European Research Council (ERC) under the



European Union's Horizon 2020 research and innovation program (IVORI, grant no. 949516). We thank Maurine Montagnat for the useful
discussion on the topic. We thank the ElmerFEM and CGAL developer teams.



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
