# Peer review of "Microstructure-based simulations of the viscous densification of snow and firn"

_EGUsphere, 2023_

## Referee Comment (RC1)

**Microstructure-based simulations of the viscous densification of snow and firn.**

**1 General comments**

This is a well-written paper presenting new and important results which will be of interest to all those interested in the densification of snow. Using information on the microstructure of snow samples from micro-CT scans, the authors calculate macroscale compaction rates under various assumptions about the relationship between strainrate and stress for the ice grains. For dense snow (firn) using an isotropic power-law with n=3 on the microscale leads to macroscale compaction rates very similar to those observed in the field, supporting the suggestion that firn behaves as a "foam" of polycrystalline ice. On the other hand, the simulated compactive viscosities for lower density alpine snow are significantly larger than the observed values, both for rounded grains and for depth hoar. This discrepancy cannot be removed by changing the power-law exponent or by using an anisotropic flow law on the microscale.

The authors note that grain-boundary sliding has been identified as a mechanism for compaction in lower density snow, but explain that simulation of this process on the microscale would require more complex numerical methods than those used in this paper. Nevertheless this is probably the next problem to tackle in this field. In the meantime this paper is a valuable contribution to the series of papers deriving various macroscale properties of snow from microphysical analysis.

**2 Specific comments**

l.325 The comment here that use of a Newtonian rheology cannot be directly motivated by an underlying physical argument seems at odds with the earlier reference to Nabarro-Herring creep (l.49) which is the physical basis for Arthern's use of a linear relationship between strain rate and stress.

**3 Technical corrections**

- l.1 The distinction between snow and firn according to density (not age) needs to be introduced here as well as in the Abstract, bearing in mind that it may not be familiar to all readers

- l.3 maybe "are still largely based on macroscale experiments" would be better?

- l.10 "firn *densification* can be reasonably well simulated"

- l.12 "contradiction"

- l.16 "firn as a foam"

- l.21 "in the cryospheric sciences"?

- l.24 it is not clear what "different layers" means here. Different depths maybe? Or different samples with the same density and/or overburden pressure but different microstructure and/or composition?

- l.25 "This situation is remarkably similar in snow"

- l.34 "*The* effective material properties" implies that all these properties can now be derived. Better to say "Effective material properties..." which only implies that some can be derived

- l.40 How about "Despite the pressing need for an accurate model,..."

- l.43 " so far only......have attempted to estimate"

- l.47 would " of the material" be better here?

- l.56 "...who considered three different ...."

- l.59 "can be simulated consistently.."

- l.60 similar to what?

- l.63 why not simply "where observed densification rates are available"?

- l.64 "computational platform as it is already established in the ice flow modelling community"

- l.71 Do you mean " it would be impossible/impractical to represent... in a snow or firn model"?

- l.76 "modelling purposes a macroscopic constitutive law .... is required. Here f is a function...."

- l.95 The colon product will be unfamiliar to many readers - explain or avoid?

- l.139 "in order to compare our simulations with independent estimates" and "These estimates are used for the comparison" seem to be saying the same thing

- l.145 RG and DH need to be defined here

- l.153 "acceleration due to gravity"

- l.155 "data include"

- l.159 "in order to estimate the uncertainty"?

- l.159 "a total... was" or " 25 time series were"

- l.178 "B54 core was drilled"

- l.180 "profile"

- l.181 ".. density profile represents a steady-state"

- l.184 Maybe use a variable like $\tau$ to represent age?

- l.185 "As in the case.."

- l.187 "weighing"

- l.188 "in a 1 m core"

- l.192 "still fluctuate"

- l.193 "As with the alpine case..."

- l.195 "envelopes"

- l.202 "The goal of these simulations was..." Similarly in l.206 and l.209 "was" is better than "is" since the rest of the description of the method is in the past tense

- l.223 "ice sheet modelling"

- l.260 Eq. 3

- l.267 "evaluated as..."

- l.271 flattened or flat

- l.284 " Several works in the literature have proposed" or maybe "Several authors propose"

- l.287 "subsequent work by..."

- l.290 described by Glen's law? known fluidity values?

- l.292 "ice fluidities"

- l.302 "who reached a similar conclusion"?

- l.304 "Moreover, Fig. 6 shows that while..."

- l.326 "increase our understanding"?

- l.328 "following Wautier et al."

- l.337 "confirmed from the simulated stress distribution..."

- l.343 "whether linear or non-linear"

- l.353 " driven by a transition in density"

- l.396 space missing after "dislocation creep"

- l.402 " ice rheology based on ..."?

- l.404 " In this way, the difference in scales..."

---

## Referee Comment (RC2)

Manuscript ID egusphere-2023-1928
**Microstructure-based simulations of the viscous densification of snow and firn.**

Kévin Fourteau, Johannes Freitag, Mika Malinen and Henning Löwe

February 7, 2024

In this paper, the authors present significant contributions to the homogenization of the viscous behavior of snow and firn. They perform finite element simulations of the mechanical behavior of snow and firn in oedometer conditions based on X-ray micro-tomography images. They compare the homogenized viscous behavior to experimental results to back analyze the micro origin of the viscous behavior. In particular, they discuss in details the modeling of the ice matrix as a poly-crystal in case of firn (isotropic behavior) and as a mono-crystal in case of snow (anisotropic behavior). This is done by considering a sensitivity analysis on different ice rheologies.

The paper is well written, easy to follow with a rather clear three dimensional formulation of the viscous behavior of ice and snow. I recommend publication subjected to the minor following comments.

1. In the simulations of the mechanical response of snow and firn samples, did the authors model the transient elasto-visco-plastic regime? How did they isolate the visous response?

2. Does the local anisotropy of the ice behavior reflect on the macroscopic behavior, or does the local fluctuations in the directions of ice anisotropy cancel out at the macroscale?

3. Complementary to the given reference (Tsuda et al. 2010), I would like to underline a few theoretical references showing that the exponent $n$ of the viscous behavior of a porous material is preserved in the up-scaling process. The authors could also refer to Auriault et al., 1992; Suquet, 1993; Orgéas et al., 2007.
   *Auriault, J.-L., Bouvard, D., Dellis, C., and Lafer, M.: Modeling of hot compaction of metal powder by homogenization, Mech. Mater., 13, 247–275, 1992.*
   *Suquet, P.: Overall potentials and extremal surfaces of power law orideally plastic composites, J. Mech. Phys. Solids, 41, 981–1002, 1993.*
   *Orgéas, L., Geindreau, C., Auriault, J.-L., and Bloch, J.-F.: Upscaling the flow of generalised Newtonian fluids through anisotropic porous media, J. Non-Newton. Fluid, 145, 15–29, 2007.*

4. As a curiosity, the authors could include some explanations on how the ice matrix switches from mono to poly-crystals when snow transforms into firn.

5. In addition to the given references, the anisotropic formulation of the viscous behavior of the ice behavior (which relies on the form of the fourth order tensor $a$) could be included explicitly in the text to have a self-supporting paper. In the mono-crystal model, what are the conditions applied on the interfaces between two crystals?

6. When referring to the segmentation of the ice matrix into mono-crystals (l.215), the authors could refer more explicitly to the images obtained using diffraction X-ray micro-tomography.

7. Maybe the authors could consider moving Section 3.3 "testing the finite element setup" in an appendix.

---

## Author Response (AR1)

Dear Guillaume Chambon,

Thank you for editing this manuscript and for your comments. Please find below our point by point response to your report and to the two reviews of Elizabeth Morris and Antoine Wautier. In each case, the comments are shown in blue and our responses in black below. Proposed modifications to the manuscript are shown in highlighted yellow.

Please note that besides the remarks pointed out during the review we have also corrected a typo **L263**, where the unit of the fluidity a has been corrected from $Pa^{-3}\ s^{-3}$ to $Pa^{-3}\ s^{-1}$.

A track-change version of the manuscript is available below, with removed text in red strike-out and added text in underlined blue.

The paper reports on microstructure-based finite-element simulations of snow and firn that aim at testing the capability to reproduce macroscopic compaction rates under the assumption that the main deformation mechanism is the viscoplastic creep of ice. The originality of the study lies in the direct cross-comparisons between numerical results and independent, field-derived densification data. It is shown that firn compaction is correctly captured by using the classical Glen's law for ice at the microstructure level. In contrast, the compactive viscosity of snow is significantly overestimated by the simulations, and this regardless of the considered microstructure type (rounded grains or depth hoar). Varying the n exponent of the rheology, or considering an anisotropic constitutive law, do not resolve this discrepancy. These results suggest that alternative deformation mechanism, such as grain boundary sliding, may play a significant role.

The manuscript meets the criteria to be put forward into the interactive discussion and sent to referees. It is well-written and clearly structured. The methods are correctly explained and, overall, the conclusions are convincing, providing new insights into the mechanical behaviour of snow. That being said, I think that the contribution could be further strengthened by a few complementary results and discussions that would probably require only a minimal amount of additional work. In particular, and depending also on the comments that will be made by reviewers, I suggest that the authors exploit the revision phase to consider addressing the following issues:

\* Would it be possible to infer the n exponent directly from the field data?
The non-linear exponent could indeed be inferred from experimental data. The use of field data might be difficult, as the external conditions are not controlled and it might be complicated to disentangle the effect of n from those of temperature, overburden and microstructure. However, attempts have been carried out to measure n through controlled laboratory experiments for both snow (Scappoza et al., 2003, Sundu et al., 2024) and firn (Li and Baker, 2022).
These observations support the notion that dense/large-grained snow and firn have a non-linear exponent consistent with polycristalline ice (n≈3.5 to 4) and that light/fine-grained snow has a lower non-linear exponent (n≈2). However, while the knowledge of the non-linear exponent helps characterizing the deformation mechanism at play in snow and firn, it is not sufficient to unequivocally determine the appropriate rheology to upscale a microstructure to its equivalent compactive viscosity.

These points will be further discussed in the manuscript.

We have completed the paragraph **L353** to indicate that a transition in the non-linear exponent has been experimentally observed and supports the notion of ice rheology transition:

*"There thus would be a transition in the ice rheology from snow, characterized by freely-deforming mono-crystals, to firn, characterized by the interaction of incompatibly oriented crystals (i.e. polycrystalline ice), as the crystals start blocking one another. This vision is consistent with experimental observations of the non-linear exponent n of snow and firn that shows a transition from n≈2 to n≈3.5-4.5 (Scapozza et al.,2003, Sundu et al., 2024). Such a transition in behavior, which is assumed to be driven by a transition in density, is widely adopted (Alley, 1987firn, Arnaud et al., 2000, Salamatin et al., 2009, Morris et al., 2022)."*

We have also completed the paragraph **L397** presenting the experimental results on the non-linear exponent:

*"The non-linear exponent of firn has recently been experimentally studied by Li and Baker (2022), which rather report a non-linear exponent of n≈4. This value is consistent with the laboratory experiments of Sundu et al. (2024) that report n≈4.4 for large-grained snow."*

We have also included **L423** a reference to the recent study of Sundu et al. (2024) in the paragraph discussing whether density alone is the relevant parameter controlling the transition in ice rheology:

*"On the other hand, our viscosity ratios also raise the question of whether the density (of snow or firn) is actually the relevant property that discerns between different deformation mechanisms (Alley, 1987, Morris et al., 2022). Our snow samples were selected to include cases (Fig.5) with virtually the same density but clear differences in the observed densification rates. Consistent with this idea, the recent study of Sundu et al. (2024) suggests that the transition in the ice matrix rheology (characterized in their study by a transition from n≈1.9 to n≈4.4) is better captured by grain-size than by density. "*

* Showing some stress – strain-rate curves obtained from simulations on the real microstructures, and discussing whether the results effectively agree with an identical n exponent at the micro and macro scales, would be interesting.

We propose to replace the simulations performed on an idealized microstructure by a set of simulations performed on an actual firn microstucture. Results confirm that the exponent n is conserved at the macroscopic scale for a real microstructure as well.

Figure 3 will be replaced with the results obtained from the firn microstucture, we will modify the text **L261** and **L266**:

*"To this end we used the microstructure obtained from one of the B34 firn samples."*

*"The corresponding compactive viscosity of this microstructure can be evaluated as $\eta=28.1\ Pa^3\ s$ (and this value of course depends on the specific choice of the non-linear exponent n and of the fluidity pre-factor a)"*

as well as the **caption of Fig 3.**:

*"Stress versus Strain rate curve characterizing the response of a firn microstructure composed of isotropic ice under compression (and assuming a fluidity prefactor $a=1\ Pa^{-3}\ s^{-1}$). The macroscopic response follows a power-law with the same exponent as the ice material, here with n=3."*

* In section 4.3, the assumption of an isotropic c-axis distribution for DH snow seems questionable. One could argue that the difference between RG and DH compactive viscosities likely stems from the presumably anisotropic texture of the latter snow type. Hence, would it be possible to consider running a few additional simulations to assess the potential effect of such texture anisotropy?

To the best of our knowledge the link between snow type (DH, RG, etc) and crystallographic texture is not always very clear. For instance, Riche et al., 2013 studied the evolution of the texture under temperature gradient metamorphism, with an evolution of the snow type from DF (close to RG) into DH, in two cold-room experiments. They report (Fig. 4, 5, and 6 of their article) initial textures with preferentially vertically orientated c-axes and which either (i) essentially stay the same, or (ii) evolve towards a weak griddle-type (c-axes in the horizontal plane) texture.

Therefore, we do not think that DH can be understood as significantly and systematically more texturally anisotropic than other snow types. Thus, it does not appear that the fact that DH is systematically reported to have a larger compactive viscosity than other snow types could directly be attributed to a more anisotropic texture.

To further investigate this point, we have performed an extra simulation of the DH sample (from 2016-02-16) with c-axes preferentially vertically orientated (roughly corresponding to the strongest fabric reported in Riche et al., 2013). We found that in this case the compactive viscosity of the DH decreased slightly, yielding a DH/RG viscosity ratio on the 2016-02-16 of 0.08+/-0.02 (versus 0.11 in the isotropic texture case). In comparison, the observed viscosity ratio is around 1.

We thank Elizabeth Morris for her review and her constructive remarks on the manuscript. Please find below our point by point response to the review. The comments of the referee are shown in blue and our corresponding responses in black below. Proposed modifications to the manuscript are provided as highlighted text with the lines corresponding the submitted manuscript.

**1 General comments**

This is a well-written paper presenting new and important results which will be of interest to all those interested in the densification of snow. Using information on the microstructure of snow samples from micro-CT scans, the authors calculate macroscale compaction rates under various assumptions about the relationship between strainrate and stress for the ice grains. For dense snow (firn) using an isotropic power-law with n=3 on the microscale leads to macroscale compaction rates very similar to those observed in the field, supporting the suggestion that firn behaves as a "foam" of polycrystalline ice. On the other hand, the simulated compactive viscosities for lower density alpine snow are significantly larger than the observed values, both for rounded grains and for depth hoar. This discrepancy cannot be removed by changing the power-law exponent or by using an anisotropic flow law on the microscale. The authors note that grain-boundary sliding has been identified as a mechanism for compaction in lower density snow, but explain that simulation of this process on the microscale would require more complex numerical methods than those used in this paper. Nevertheless this is probably the next problem to tackle in this field. In the meantime this paper is a valuable contribution to the series of papers deriving various macroscale properties of snow from microphysical analysis.

**2 Specific comments**

l.325 The comment here that use of a Newtonian rheology cannot be directly motivated by an underlying physical argument seems at odds with the earlier reference to Nabarro-Herring creep (l.49) which is the physical basis for Arthern's use of a linear relationship between strain rate and stress.

Indeed there exist creep mechanisms (such as Nabarro-Herring) characterized by a linear strain rate-stress relationship. What we wanted to convey in the article is that our exploration of rheologies besides n=3 is not motivated by the existence of underlying deformation mechanisms (for instance N-H) but rather as an exploration of homogenous isotropic deformation law (and this independent of the existence of physical mechanisms that could justify such a form).

In order to better explain this point (and taking into account the technical comments below) we propose to modify the manuscript **L324**:

*"Macroscopic Newtonian rheologies are also used in low-density firn studies (Schultz et al, 2022). Therefore, exploring rheologies besides Glen's law, with related numerical experiments, would benefit our understanding of homogenization in snow compaction. Accordingly, we conducted a sensitivity analysis on microscopic, isotropic constitutive laws using different exponents (n=1, n=2, and n=4 in our case) following Wautier et al., (2017). We note that while some of these rheologies could be justified based on mechanisms of ice deformation (such as the Nabarro-Herring creep resulting in n=1; Herring, 1950, Arthern et al., 2010), our analysis of n≠3 was not conducted with specific physical mechanisms in mind. Rather, our motivation is to determine if an isotropic deformation law could explain snow compaction, independently of a specific underlying physical mechanism."*

**3 Technical corrections**

• l.1 The distinction between snow and firn according to density (not age) needs to be introduced here as well as in the Abstract, bearing in mind that it may not be familiar to all readers

We have moved the distinction between snow and firn in terms of density directly in the first sentence of the abstract **L1**:

*"Accurate models for the viscous densification of snow (understood here as density below 550 kg m$^{-3}$) and firn (density above 550 kg m$^{-3}$) under mechanical stress are of primary importance for various applications, including avalanche prediction and the interpretation of ice cores."*

• l.3 maybe "are still largely based on macroscale experiments" would be better?

We think that the word "empirical" is more suited than "macroscale experiments" here as experiments could be understood as controlled conditions (while field observations are also used to adjust models). Moreover, the word "empirical" implies that the underlying theory/understanding behind snow/firn compaction is still to be refined.

Thus, if the referee and editor agree, we propose to keep the manuscript as such.

• l.10 "firn densification can be reasonably well simulated"
• l.12 "contradiction"
• l.16 "firn as a foam"

We will reformulate the manuscript following these suggestions.

• l.21 "in the cryospheric sciences"?

We used "cryospheric sciences" to span both snow sciences and glaciology, as they are sometimes considered as two separate fields. To clarify the sentence, we propose to simply remove "in cryospheric sciences" **L21**:

*"Accurate prediction of the rate of the compaction is of primary importance for various applications. For instance, [...]"*

• l.24 it is not clear what "different layers" means here. Different depths maybe? Or different samples with the same density and/or overburden pressure but different microstructure and/or composition?

We meant the second, i.e. that layers with similar density and understand similar overburden stress can show clearly different compaction rates. We propose to reformulate **L24** to:

*"However, observed variations in the densification rate of different layers with similar density and subjected to similar overburden stress still lack a conclusive explanation in view of either microstructural or compositional origins (Hörhold et al., 2012, Fujita et al., 2016)."*

• l.25 "This situation is remarkably similar in snow"
• l.34 "The effective material properties" implies that all these properties can now be derived. Better to say "Effective material properties..." which only implies that some can be derived
• l.40 How about "Despite the pressing need for an accurate model,…"
• l.43 " so far only......have attempted to estimate"

We will reformulate the manuscript following these suggestions.

• l.47 would " of the material" be better here?

We propose to reformulate **L47** to:

*"[…] the dominating mechanism(s) driving the mechanical deformation of the ice material at the micro-scale."*

• l.56 "...who considered three different …."
• l.59 "can be simulated consistently.."

We will reformulate the manuscript following these suggestions.

• l.60 similar to what?

By "similar" we meant common between the firn and snow samples. We will rephrase **L60** replacing the word "similar" with "common".

• l.63 why not simply "where observed densification rates are available"?

We propose to rephrase **L63** to:

*"[…] where observed densification rates are also available."*

• l.64 "computational platform as it is already established in the ice flow modelling community"

We will reformulate following this suggestion.

• l.71 Do you mean " it would be impossible/impractical to represent... in a snow or firn model"?

We propose to reformulate the sentence **L71** to:

*"In snowpack and firn models, it would be impossible to explicitly represent the 3D microstructure of a whole snowpack or firn column."*

• l.76 "modelling purposes a macroscopic constitutive law .... is required. Here f is a function…."

We will reformulate **L76** to:

*"For snow or firn modelling purposes, a macroscopic constitutive law $\dot{E} = f(\Sigma)$ is required. Here, f is a function [...]"*

• l.95 The colon product will be unfamiliar to many readers - explain or avoid?

We will provide the definition of the double dot product between a fourth and second order tensor using index notation **L99**:

*"The double product **a:s** yields a second-order tensor whose $ij^{th}$ component is $\sum_{kl} a_{ijkl} s_{kl}$."*

• l.139 "in order to compare our simulations with independent estimates" and "These estimates are used for the comparison" seem to be saying the same thing

Indeed. We propose to remove the second part "These estimates are used for the comparison".

• l.145 RG and DH need to be defined here

We will specify that RG and DH refers to Rounded Grains and Depth Hoar **L145**:

*"Four snow layers have been carefully tracked and measured with several instruments over the entire season, including a Rounded Grains (RG) snow layer and a Depth Hoar (DH) snow layer (following the classification of Fierz et al., 2009)."*

• l.153 "acceleration due to gravity"
• l.155 "data include"
• l.159 "in order to estimate the uncertainty"?

• l.159 "a total... was" or " 25 time series were"
• l.178 "B54 core was drilled"
• l.180 "profile"
• l.181 ".. density profile represents a steady-state"
• l.184 Maybe use a variable like τ to represent age?
• l.185 "As in the case.."
• l.187 "weighing"
• l.188 "in a 1 m core"
• l.192 "still fluctuate"
• l.193 "As with the alpine case..."
• l.195 "envelopes"
• l.202 "The goal of these simulations was..." Similarly in l.206 and l.209 "was" is better than "is" since the rest of the description of the method is in the past tense
• l.223 "ice sheet modelling"
• l.260 Eq. 3
• l.267 "evaluated as..."
• l.271 flattened or flat
• l.284 " Several works in the literature have proposed" or maybe "Several authors propose"
• l.287 "subsequent work by..."
We will reformulate the manuscript following these suggestions.

• l.290 described by Glen's law? known fluidity values?
We propose to reformulate **L290** to:
*"The viscoplastic deformation of polycrystalline ice is nowadays reasonably well understood and usually described by Glen's law, an isotropic power-law rheology with n=3 and known fluidity values depending on the temperature of the ice (Schulson and Duval, 2009, Cuffey and Paterson, 2010)."*

• l.292 "ice fluidities"
• l.302 "who reached a similar conclusion"?
• l.304 "Moreover, Fig. 6 shows that while..."
We will reformulate the manuscript following these suggestions.

• l.326 "increase our understanding"?
We propose to reformulate **L326** to:
"[…] would benefit our understanding [...]"

• l.328 "following Wautier et al."
• l.337 "confirmed from the simulated stress distribution..."
• l.343 "whether linear or non-linear"
• l.353 " driven by a transition in density"
• l.396 space missing after "dislocation creep"
• l.402 " ice rheology based on ..."?
• l.404 " In this way, the difference in scales…"
We will reformulate the manuscript following these suggestions.

We thank Antoine Wautier for reviewing the manuscript and his constructive remarks. Please find below our point by point response. The comments of the referee are shown in blue and our corresponding responses in black below. Proposed modifications to the manuscript are provided as highlighted text with the lines corresponding the submitted manuscript.

In this paper, the authors present significant contributions to the homogenization of the viscous behavior of snow and firn. They perform finite element simulations of the mechanical behavior of snow and firn in oedometer conditions based on X-ray micro-tomography images. They compare the homogenized viscous behavior to experimental results to back analyze the micro origin of the viscous behavior. In particular, they discuss in details the modeling of the ice matrix as a poly-crystal in case of firn (isotropic behavior) and as a mono-crystal in case of snow (anisotropic behavior). This is done by considering a sensitivity analysis on different ice rheologies.
The paper is well written, easy to follow with a rather clear three dimensional formulation of the viscous behavior of ice and snow. I recommend publication subjected to the minor following comments.

1. In the simulations of the mechanical response of snow and firn samples, did the authors model the transient elasto-visco-plastic regime? How did they isolate the visous response?
In this article we consider a purely visco-plastic material, without an elastic component. This way we directly obtain the steady-state macroscopic viscous response without the need to model a potential transient elastic response.

The fact that we do not need to isolate the viscous response will be specified in the "Finite element solution" Section **L232**:
"*As we consider a purely viscous material without elasticity, we do not need to isolate the viscous response from an elastic part (as for instance done in Wautier et al., 2017).*"

2. Does the local anisotropy of the ice behavior reflect on the macroscopic behavior, or does the local fluctuations in the directions of ice anisotropy cancel out at the macroscale?
In our simulation framework we have only considered the vertical response, as it is the relevant direction for natural snowpacks and firn columns. We thus did not attempt to estimate the mechanical anisotropy of the material. This limitation will be mentioned in the manuscript **L117**:
"*Also, this study is limited to the investigation of vertical compaction in snowpacks/firn columns and does not consider other directions of deformation (e.g. lateral compaction). Therefore, we do not quantify the potential anisotropy of the compactive viscosity.*"

As hinted by the referee, since we are considering isotropic crystallographic textures in this article, the fluctuations due to the crystallographic orientations should cancel out as there is no preferential direction for the c-axis. The study of a texture-induced anisotropy, and its interaction with the anisotropic microstructure of snow and firn is an interesting path of investigation, with applications in the case where snow and firn are subjected to more than pure oedometric vertical strain. We however feel that this path is beyond the scope of our article as it would first require to determine the relevant ice deformation mechanism in snow before performing a large number of specific numerical simulations. We propose to mention in the text that the study of the interaction between textural and structural anisotropies could be worthy of investigation in the future **L422**:
"*A deeper understanding of the influence of the snow texture on its mechanical properties would enable the study of the interaction between structural and textural anisotropies.*"

3. Complementary to the given reference (Tsuda et al. 2010), I would like to underline a few theoretical references showing that the exponent n of the viscous behavior of a porous material is preserved in the up-scaling process. The authors could also refer to Auriault et al., 1992; Suquet, 1993; Orgéas et al., 2007.

We thank the referee for providing these references. They will be added to the manuscript **L104**.

4. As a curiosity, the authors could include some explanations on how the ice matrix switches from mono to poly-crystals when snow transforms into firn.

We propose to extend the paragraph **L353** to specify how and why the ice rheology transitions from mono to polycristalline:

*"There thus would be a transition in the ice rheology from snow, characterized by freely-deforming mono-crystals, to firn, characterized by the interaction of incompatibly oriented crystals (i.e. polycrystalline ice), as the microstructure becomes denser and the crystals start blocking one another."*

5. In addition to the given references, the anisotropic formulation of the viscous behavior of the ice behavior (which relies on the form of the fourth order tensor a) could be included explicitly in the text to have a self-supporting paper. In the mono-crystal model, what are the conditions applied on the interfaces between two crystals?

We will move the anisotropic formulation Appendix into the main part of the paper, in Section 3.2.2 "Finite element solution".

For the interface between two crystals we use the "natural" condition that directly follow from the FEM formulation, i.e. continuity of displacement rates. These corresponds to the condition found in a homogeneous material.

We note that it is because of this continuity that we cannot naturally model interface effects, such as grain boundary sliding. The continuity condition between crystals will be mentioned in the text **L253**:

*"In the anisotropic simulations, the condition at the interface between monocrystals is characterized by displacement rates continuity."*

6. When referring to the segmentation of the ice matrix into mono-crystals (l.215), the authors could refer more explicitly to the images obtained using diffraction X-ray micro-tomography.

We will mention that the crystal segmentation could be experimentally measured using X-ray diffraction tomography. If applicable to sufficiently large samples, this would remove the uncertainty associated with the geometrical segmentation used in our study.

We will add **L221** that the texture of our snow sample could potentially be experimentally measured though XDT:

*"We note that while such technique was not available for our study, the crystallographic orientation in snow could also be experimental determined through X-ray diffraction tomography (Roscoat et al., 2011, Reischig et al., 2013, Granger et al., 2021)."*

We will also mention it when discussing the need for texture measurments in Section 4.4 **L420**:

*"Concurrent measurements of the texture are then unavoidable, either through snow thin-sections (Riche et al., 2013, Montagnat et al., 2020) or through X-ray diffraction tomography (Roscoat et al., 2011, Reischig et al., 2013, Granger et al., 2021)."*

7. Maybe the authors could consider moving Section 3.3 "testing the finite element setup" in an appendix.
If the editor and referee agree, we would prefer to keep Section 3.3 in the main part of the manuscript, as it also illustrates some of the points presented in the Theoritical background Section (the preservation of the non-linear exponent and the influence of an anisotropic ice material). Also, as the appendix of the previous manuscript was moved to the main part of the article, keeping Section 3.3 in the main part of the manuscript would allow a single streamline article.

[revised manuscript text omitted]